# Roles and Regulation of BCL-xL in Hematological Malignancies

**DOI:** 10.3390/ijms23042193

**Published:** 2022-02-16

**Authors:** Mario Morales-Martínez, Mario I. Vega

**Affiliations:** 1Molecular Signal Pathway in Cancer Laboratory, UIMEO, Oncology Hospital, Siglo XXI National Medical Center, IMSS, México City 06720, Mexico; ixnergal@gmail.com; 2Department of Medicine, Hematology-Oncology Division, Greater Los Angeles VA Healthcare Center, UCLA Medical Center, Jonsson Comprehensive Cancer Center, Los Angeles, CA 90073, USA

**Keywords:** BCL-xL, hematological malignances, non-Hodgkin’s lymphoma

## Abstract

Members of the Bcl-2 family are proteins that play an essential role in the regulation of apoptosis, a crucial process in development and normal physiology in multicellular organisms. The essential mechanism of this family of proteins is given by the role of pro-survival proteins, which inhibit apoptosis by their direct binding with their counterpart, the effector proteins of apoptosis. This family of proteins was named after the typical member Bcl-2, which was named for its discovery and abnormal expression in B-cell lymphomas. Subsequently, the structure of one of its members BCL-xL was described, which allowed one to understand much of the molecular mechanism of this family. Due to its role of BCL-xL in the regulation of cell survival and proliferation, it has been of great interest in its study. Due to this, it is important to research its role regarding the development and progression of human malignancies, especially in hematologic malignancies. Due to its variation in expression in cancer, it has been suggested that BCL-xL can or cannot play a role in cancer depending on the cellular or tissue context. This review discusses recent advances in its transcriptional regulation of BCL-xL, as well as the advances regarding the activities of BCL-xL in hematological malignancies, its possible role as a biomarker, and its possible clinical relevance in these malignancies.

## 1. Introduction

Hematological malignancies are defined as a set of neoplastic conditions that occur especially in hematopoietic tissue and in lymphoid tissue such as bone marrow and lymph nodes, whose representative diseases include leukemias, lymphomas, and multiple myeloma [1]. In these malignancies as in others, an important participation of the BCL-xL protein has been reported, since in lymphoma it has been reported that 80% of these have an overexpression of the protein [2], similar to what has been observed in leukemia [3]. For multiple myeloma, there are few reports. However, based on a published study, the expression has been associated with an increase in the malignancy of the disease [4]. That is why the BCL-2 family of proteins, and specifically BCL-xL, has high importance as therapeutic targets for the treatment of cancer given their function as inhibitors of apoptosis and resistance to therapy [5].

The BCL-2 family is made up of several proteins which are classified according to their function derived from their structure in anti-apoptotic proteins such as the BCL-2, BCL-xL, BCL-W, A1, MCL1, BCL-RAMBO, BCL-B, and BCL-G proteins, the pro-apoptotic proteins represented by BAX, BAK, BID, and BOK while the members are considered pro-apoptotic effectors and the BH3-only proteins as BIM, BAD, BIK, NOXA, HRK, and PUMA, (Figure 1) [6].

Due to its participation in the regulation of apoptosis, a process whose evasion is directly related to the generation of malignant processes [7], alteration in the expression of the members of the BCL-2 family has been reported as fundamental in the generation and promotion of cancer [8]. Therefore, the study of the BCL-2 protein family is fundamental for the understanding and potential treatment of cancer. One of the most important members of the BCL-2 family is BCL-xL, a protein described in 1993 [9], and whose mechanism is the regulation of the permeability of the mitochondrial membrane and the release of cytochrome C [10]. Likewise, it has been reported that the BCL-xL knockout results in severe defects in erythropoiesis and neuronal development in mice. [11].

BCL-xL also called BCL2L1, is located on chromosome 20 (20q11.21), it is a consequence of the alternative splicing of the human BCL-X gene, which generates the BCL-xs isoforms (591 bp) with three exons and the BCL isoform -xL (780bp) with four exons [12]. BCL-xL protein is 233 aa long and is made up of a total of eight alpha-helices (Figure 2) [13].

In the context of cancer, BCL-xL is normally over-expressed compared to healthy tissue [14]. In addition to the role of BCL-xL in tumor suppression, the expression of this protein has been related to other malignant processes such as invasive capacity [15], induction of transitional mesenchymal epithelium, and metastasis independent of its antiapoptotic activity [16]. This is likely due to the targets on which it acts, among which are TGF-β [16], YY1 [17], and KLF4 [18], all of which have been closely related to metastasis, EMT and dedifferentiation, respectively. In some malignancies, the participation of BCL-xL in the regulation of other proteins has been demonstrated. For example, in glioblastoma the participation of BCL-xL was demonstrated as an inducer of interleukin-8 expression through the regulation of NF-κB [19]. However, in melanoma cells, the induction of interleukin-8 through BCL-xL was related to aggressiveness, cell spread, and angiogenesis. [20,21]. Additionally, it has been proposed that in glioblastoma and melanoma BCL-xL contributes and promotes stemness and aggressiveness [14]. The ability of BCL-xL to confer resistance to chemotherapy has been established in various cell models. This ability to prevent apoptosis in response to DNA damage by chemotherapy and cell cycle arrest could contribute to the accumulation of chromosomal aberrations [5].

## 2. BCL-xL in Hematological Malignancies

As in various types of cancer, in hematological malignancies an overexpression of BCL-xL has been reported [22] which has given rise to the development of therapies focused on the resistance generated by this overexpression. Such is the case of Venetoclax (ABT-199) [23] and Navitoclax (ABT-263) [24], a small molecule inhibitor directed against BCL-2, clinically approved for use in ALL, CLL, and AML [23]. Also, another new inhibitor is under study such as that reported by Yi Zhu et al., who propose A1331852 and A1155463, specific inhibitors of BCL-xL, with a probable lower hematological toxicity [25].

Recently, alternative therapies have been proposed based on the use of peptides derived from the BH3 domain of the pro-apoptotic protein Bax, which have the ability to antagonize the anti-apoptotic effect of the BCL-2 family of proteins [26]. In this context, various strategies have been proposed. One of them consists of the expression and release of a Bax BH3 peptide permeable to cells on the surface of *Salmonella enterica* serovar *Typhimurium* SL3261 through the MisL autotransporter system, which demonstrated significant antitumor activity in non-Hodgkin lymphoma cell lines as well as in a murine model [27].

### 2.1. BCL-xL in Non-Hodgkin Lymphoma

In order to establish an overview of the expression of BCL-xL in various hematological malignancies, a bioinformatic analysis of expression was performed in various types of hematological malignancies, among which NHL subtypes DLBCL, FL, chronic lymphocytic leukemia, and multiple myeloma stand out. For this analysis, the ONCOMINE database was used [28], as well as the data set “Basso lymphoma” [29]. Results of this analysis show that, compared to B lymphocytes as a control, the NHL subtypes Burkitt’s lymphoma, DLBCL, and FL show a trend towards higher expression of BCL-xL, while, on the other hand, multiple myeloma and CLL show an expression similar to the control. (Figure 3A). Similar results were obtained after the analysis of the dataset. “Storz Lymphoma” [30], in which it can be observed that, compared to tonsils used as a control, the expression of BCL-xL in cutaneous follicular lymphoma (CFL), DLBCL and marginal zone B cell lymphoma is increased (Figure 3B). These results suggest the important participation of this protein in malignancies described here, with a special participation in lymphomas. Thus, this protein in the context of hematological malignancies requires further study.

The expression of BCL-2 in lymphomas has been widely reported, the t (14:18) translocation generates a constitutive expression of BCL-2 derived from the juxtaposition of BCL-2 with the immunoglobulin heavy chain promoter in follicular lymphoma [31], and in 30% diffuse large B-cell lymphoma [32].

In lymphoma, it has been reported that the expression of BCL-xL is commonly expressed in malignant cells of various types of Hodgkin’s lymphoma, as well as in non-Hodgkin’s lymphomas [2]. In particular, the overexpression of BCL-xL has been reported in 80% of all non-Hodgkin’s lymphomas. Therefore, it represents an important molecular target for the treatment of the disease.

Obatoclax is a small-molecule Bcl-2 antagonist that binds the hydrophobic groove in Bcl-2, BCL-xL and Mcl-1 [33] and mimics pro-apoptotic BH3-only proteins, such as Bad, Bim, Noxa, and Puma from sequestering the multidomain pro-apoptotic proteins (Bak and Bax). Obatoclax treatment of NHL cell lines show that it induces inhibition of NF-kB, which induces an inhibition of the expression of Mcl-1 and BCL-xL. In conjunction with the inhibition of the negative regulator of DR5, YY1 leads sensitizing these cell lines to TRAIL [34].

There are few studies that evaluate the expression of BCL-xL in lymphoma, so using the ONCOMINE data platform [28], we analyzed the expression of BCL-xL (BCL2L1) in the data set called Compagno Lymphoma [35], in which comparing memory B lymphocytes against diffuse large B-cell lymphoma of germinal center, diffuse large B-cell lymphoma of activated B lymphocytes, and diffuse B-cell lymphoma and follicular lymphoma, we discovered an important higher expression in malignancies compared to healthy control (Figure 4).

### 2.2. BCL-xL in Leukemia

Similar as reported in non-Hodgkin’s lymphoma, in leukemia the expression of BCL-xL is increased (3). Additionally, it has been related with apoptosis inhibition, resulting in chemoresistance [36]. Specifically, high levels of BCL-xL expression in cell lines have been reported to protect against cytotoxic agents [37]. Similar to that reported in follicular lymphoma patients, acute myeloid leukemia and acute promyelocytic leukemia patients, has been reported with t (11; 14) (q32; q21) translocation with an overexpression of BCL-2, which has been related as an important factor in the pathogenesis of the disease, also associated with a poor prognosis [38]. Interestingly, the group of Jie Zou et al., proposed that hypoxia had been related to an increase in the expression of Bcl-2 and BCL-xL, allowing a marked decrease in apoptosis mediated by the activity of caspases [39]. In patients with acute myeloid leukemia, BCL-xL overexpression was related to resistance to apoptosis, poor response to chemotherapy in addition to a poor prognosis, with leukemic infiltrations, and high white blood cell counts [40]. While in conjunction with BCL-2 a possible implication in the risk of relapse has been reported. That is why the search for specific BCL-xL inhibitors has generated important results in the treatment of leukemia. Navitoclax, a specific inhibitor of BCL-xL and Bcl-2, has demonstrated clinical efficacy and antitumor activity [41]. In this context, leukemia therapies aimed at promoting apoptosis are one of the most promising in the treatment of leukemia [42].

On the other hand, it has been pointed out that the inhibition of the overexpression of BCL-xL is not enough for an effective treatment [40], therefore the use of BCL-xL inhibitors has been combined with chemotherapy. An example is the administration of INK128, an inhibitor of mTORC1 and mTORC2 in conjunction with ABT-737, an antagonist of Bcl-2 and BCL-xL, the co-administration of which resulted in a potent in vitro and in vivo anti-leukemic activity [43].

In order to understand the participation of BCL-xL in leukemia, we carried out a bioinformatic analysis with the help of the ONCOMINE database, using the public data reported by Anderson and Co. [44], which allowed to evaluate the expression of this protein in AML, B-ALL and T-ALL. As observed, there is a higher expression of BCL-xL in malignancy cells compared to the bone marrow control (Figure 5).

### 2.3. BCL-xL in Multiple Myeloma

In multiple myeloma, the expression of BCL-xL has been reported in murine myeloma plasma cells. In humans, BCL-xL is expressed in normal lymph node plasma cells, malignant plasma cells, and in various myeloma cell lines. Interestingly, the correlation of BCL-xL expression with low rates of response to treatment was reported, this being 83–87% in BCL-xL negative patients and 20–31% in patients with BCL-xL expression. Additionally, in dox-40 cells, which have an overexpression of BCL-xL, a generalized resistance to apoptosis induced by different cytotoxic agents was demonstrated. [45]. The participation of BCL-xL in resistance to treatment has been evidenced in multiple myeloma cells that co-express Bcl-2 and BCL-xL, which were resistant to treatment with venetoclax but were sensitive to treatment with the specific inhibitor of BCL-xL A-1155463. Meanwhile in MM xenograft models, the co-expression of BCL-xL, Mcl-1 and Bcl-2 were more sensitive to the combination of a selective inhibitor of BCL-xL (A-1331852) but not with venetoclax treatment as compared to monotherapies. This suggests that BCL-xL is an important factor in resistance to therapy [46]. Therefore, BCL-xL has been proposed as a prognostic factor and potential biomarker of chemoresistance. Currently there are therapies directed against the BCL-2 family in MM, such as antisense oligonucleotides (which mimics BH3-only) and small molecules with the ability to inhibit BCL-2 expression. G3139 or Oblimersen which is an antisense oligonucleotide complementary to the messenger RNA of BCL-2 [47]. Oblimersen treatment has shown, in a study phase two, an increase in apoptosis in cells of patients with MM [48]. On the other hand, and through techniques such as high-performance nuclear magnetic resonance, they have allowed the development of small molecules such as ABT-737 [49], a small molecule mimic of BH3-only, and other analogues such as ABT-263 (navitoclax) [50]. They are capable of inducing apoptosis. However, they have physiological limitations. On the other hand, other analogues have made it possible to counteract adverse effects such as ABT-199 venetoclax and GX015-070 Obatoclax [51]. Even treatment with plant derivatives such as Gossypol has been shown to interfere with the antiapoptotic functions of BCL-2 in cell lines [48].

As has been described, the participation of Bcl-2 has been widely reported and related to an important participation in the pathogenesis of MM, however, specifically, BCL-xL has been little studied. That is why we performed a bioinformatic analysis, looking for the expression of BCL-xL in multiple myeloma samples, based on public data from the Oncomine database [28]. Using the data set reported by Chapman and Co. [4], BCL-xL expression in MGUS was compared with SM and MM, very interestingly a significant increase in the expression is found that correlates with the increased risk of malignancy, with MM being the pathology with the highest expression of BCL-xL. (Figure 6).

## 3. Regulation of BCL-xL

### 3.1. BCL-xL Regulation by Transcription Factors

Due to its importance in apoptosis, the regulation of BCL-xL has been studied. However in the context of hematological malignancies its regulation is still not very clear and transcription factors, can play a fundamental role. Such is the case of the inducible hypoxia factor HIF-1, which by reporter gene assays, chromatin immunoprecipitation, and EMSA was shown that it is capable of regulating the expression of BCL-xL by binding to responding elements in the BCL-xL promoter in prostate cancer cells [52]. Interestingly, it has been reported that in myeloid progenitors the regulation of BCL-xL is given by the JAK signaling pathway and is independent of STAT3, PI3K, and RAS [53]. On the other hand, in multiple myeloma cells, an important expression of Mcl-1 and BCL-xL was found and it was shown that both proteins are regulated by interleukin 6 [54]. It was recently reported that STAT, Rel/NFκB as well as the ETS family of transcription factors are capable of regulating BCL-xL in a model of erythroleukemic cell lines and progenitor cell lines [55].

According to the public repository The Signaling Network Open Resource 2.0 (SIGNOR 2.0), in homo sapiens there are five possible transcriptional regulators of BCL2L1, which are negative regulators SH2B3 and CD79A, while positive regulators are CD27, CD40 and CREB1, as well as STAT5A and FLT3 have been associated with a regulation of BCL2L1 in mice (Figure 7) [56].

The motif search tool contained in the Eucariotic Promoter database (EPD) (https://epd.epfl.ch) [Date accessed: 10 January 2022] [57], for the BCL2L1-1 promoter, indicates representative transcription factors with at least one potential site determined by algorithm (*p* = 0.001) (Table 1). From this analysis, the transcription factors shown in Figure 8 were found with at least one potential binding site for each transcription factor predicted in the BCL-xL promoter sequence from −2000 to 100 pb, obtained from (EPD) (https://epd.epfl.ch) [Date accessed: 10 January 2022] (Figure 8).

Our research group has reported that YY1 is capable of regulating the expression of BCL-xL [58], in addition, YY1 is capable of regulating KLF4 [59], and according to the analysis of potential binding sites in the promoter (Table 1), KLF4 has the potential to regulate BCL-xL. Thus, it has also been described that HIF-1 is capable of regulating BCL-xL and YY1 contributes to stabilizing HIF-1 [60]. It has recently been reported that miR-7 plays a fundamental role in the regulation of YY1 and KLF4 in NHL [61], additionally, other authors have reported that this microRNA is capable of regulating the expression of BCL-xL and Bcl-2 [62]. Therefore, this protein could represent an important regulatory environment in the context of hematological malignancies and specifically lymphoma, which is why it should be studied (Figure 9).

### 3.2. MicroRNAs and BCL-xL

MicroRNAs are short-stranded RNAs 19 to 24 nucleotides [63], that have gained great importance due to their participation in the regulation of various biological processes such as apoptosis, cell differentiation, hematopoiesis and cancer, among others, in which the mature microRNA is capable of regulating the expression of various target proteins by binding to your region 3’UTR [64].

In the context of hematological malignancies, the first report of miRNAs involved in the pathogenesis of a disease were miRNAs 15 and 16, which are encoded in the chromosomal region 13q14 that is normally deleted in CLL. These microRNAs negatively regulate Bcl-2, antiapoptotic proteins [65]. Additionally, the aberrant expression of several microRNAs has been established (Table 2), which could be related to the prognosis or diagnosis of each disease [66], However, various studies have made it possible to elucidate the molecular mechanism by which the disease occurs or some characteristic of it.

In the context of hematological malignancies, leukemia is one of the malignancies in which the greatest number of studies on microRNAs have been reported due to its involvement from hematopoiesis and leukemogenesis [67]. As previously mentioned, the first cluster of miRNAs involved in a cancer-related process was reported in leukemia including the miR-15a and miR-16-1 clusters [67]. From that, several miRNAs have been described, such as miR-155, miR-222, and miR-34, with high expression, while others such as miR-221, miR-223, and miR-26a (among others) have a low expression which has been related to leukemogenesis [68].

As described in YY1 in the control of pathogenesis and drug resistance of cancer [69], several microRNAs have been reported as part of a possible panel of micro-markers that allow differentiating between healthy and disease, as well as between the different subtypes of leukemia (Table 3).

Studies have been carried out particularly in NHL that allowed the identification of lymphomiRs involved in various processes that contribute to lymphomagenesis, as well as their target proteins, in order to establish their role in NHL malignancy patterns (Table 4).

Additionally, various studies using tools such as RNA seq and expression microarrays have made it possible to establish expression patterns by comparing healthy tissue against tissue from different NHL samples (Table 5), which have allowed the use of lymphomiRs expression patterns as molecular tools for the identification of the different types and subtypes of lymphoma [69]

Finally, in myelodysplastic malignancies, the participation of various microRNAs, known as myelomiRs, have been reported, among which (in a very interesting way) myelomiRs is related to proliferation and migration. Chemoresistance such as in miR-150-5p and miR-125b has been reported. On the other hand, it has been reported that the expression of miR-15b, miR-16, miR-17-5p, miR-19b, miR-21, miR-22, and miR-29 allows one to differentiate patients with multiple myeloma from healthy donors [75]. Even the expression of some microRNAs has allowed the identification of genetic alterations. Such is the case of miR-135a-3p whose ectopic expression was found in patients with translocation (4:14) [76].

In addition to the establishment of expression patterns, the contribution of miRNAs in various processes typical of the various malignancies through the silencing of target genes with proven participation in the molecular mechanisms that lead to the development of malignancy has been of interest. One of the most studied cancer hallmarks is apoptosis inhibition. One of the mechanisms involved is the ectopic expression of antiapoptotic proteins [77], such is the case of BCL-xL, for which various microRNAs have been reported as responsible for its regulation.

### 3.3. LeukemiRs and LymphomiRs in the Regulation of BCL-xL

Recent studies in leukemia have reported that miR-377 is capable of negatively regulating the expression of BCL-xL in resistant ABT-199 cells, as well as in a panel of lymphoid lines of B cells and primary chronic lymphocytic leukemia. Interestingly, in the same study, it was reported that CLL patients with high expression of BCL-xL and low expression of miR-377 have an advanced tumor stage and a short survival estimation without treatment [78].

Additionally, it has been reported that in HL60 cells, the transfection of the miR-125a mimic is capable of reducing the enzymatic activity of Bcl-2 and BCL-xL [79]. Interestingly, studies in hepatocarcinoma have shown that the restoration of miR-125b expression indirectly decreases the expression of BCL-xL [80].

To find out the possible microRNAs that have been related to the expression of BCL-xL, we performed an analysis using the miRbase platform [81], available online for the prediction of microRNAs with potential binding sites in the 3’UTR region of BCL-xL. The results obtained indicated the possible participation of eight miRNAs, as well as the let-7 family (Table 6) (Figure 10).

Interestingly, these microRNAs may play a role in the types of hematological malignancy studied in this review. With a comparison between the prediction table and the expression tables previously shown, it is possible to hypothesize that microRNA-7 could play a role as a potential regulator of BCL-xL in NHL. This is consistent with previous reports in lung cancer cells in which microRNA-7 is able to regulate BCL-2 [62]. Additionally, it has been reported that in NHL cell lines, this microRNA is capable of regulating KLF4 and particularly YY1 [61], a transcription factor involved in the regulation of BCL-xL [58]. Additionally, we have reported that YY1 is capable of regulating KLF4 [59], while in KLF4/KLF5 deficient mammary cancer cells, the expression of Mcl1 and BCL-xL are reduced, suggesting a possible participation of KLF4 in the regulation of the protein [82] which is consistent with our analysis of potential binding sites for KLF4 in the BCL-xL promoter (Table 1). It has been reported that the same miRNA can regulate several proteins in the same signaling pathway, which would reinforce the potential role of miR-7 in the regulation of BCL-xL (Figure 9).

Additionally, recently reports have indicated the participation of some of the microRNAs predicted in this analysis in the regulation of BCL-xL in different diseases [83]. Decrease in miR-133 expression in osteosarcoma was related to a decrease in apoptosis, probably due to the regulation it exerts on BCL-xL and Mcl1 [84]. Finally, the let-7 family of microRNAs has been reported as a negative regulator of BCL-xL in hepatocarcinoma where tissues with a low expression of let-7c have a high expression of BCL-xL compared to those with a low expression [85].

These reports suggest that the prediction of the participation of microRNAs in the regulation of BCL-xL is potentially feasible, however, it is important to confirm if the role of miRNAs is dependent on tissue, and even on the disease, so it is necessary to evaluate the possible participation of miRNAs in the regulation of BCL-xL in hematological malignancies.

### 3.4. BCL-XL as a Potential Biomarker in Hematological Malignancies

As previously mentioned, the expression of BCL-xL in hematological malignancies has been extensively related to apoptotic processes in cell lines [10]. However, it is also feasible to relate the expression of this protein as part of a panel of biomarkers to clinical outcome. In order to establish whether there is a relationship between the expression of BCL-xL and the international prognostic index (IPI), we analyzed the expression of the protein in the dataset called “Rosenwald Lymphoma” [86], Interestingly, we found that there is a tendency to a greater expression of BCL-xL the higher the IPI. However, it does not have a statistical significance (Figure 11A). Finally, to determine if there was any relationship between the expression of BCL-xL with total survival, we analyzed the expression compared to the total survival state. For this, we used the data set reported by Lenz and Co. [87]. As can be seen in Figure 11B, there is a high expression of BCL-xL in dead patients compared to living patients. These data are consistent with what has been reported. However, in this study only one trend is observed, so more studies are required to establish the expression of BCL-xL as a marker or therapeutic target.

On the other hand, in the context of leukemia and based on the Heuser leukemia data sets [88], in ONCOMINE, we have found that the expression of BCL-xL is decreased in alive vs. dead patients (Figure 12A). Additionally we observe that data on leukemia patients obtained from the Bullinger leukemia data set [89] and patients with a complete response to treatment generally have a lower expression of BCL-XL compared to refractory disease (Figure 12B). A similar process occurs with treatment failure, in which it is observed that those patients who fail treatment have higher expression compared to responding patients in the Heuser Leukemia data set [88] (Figure 12C). In the same dataset, it can be observed that a favorable outcome is related with a lower expression of BCL-xL compared to those who had a tendency to greater expression. Therefore, it is feasible to think that BCL-xL could play a role as a potential biomarker directly related to a poor prognosis (Figure 12D). Interestingly, the analysis of the expression of BCL-xL as a function of the BCR-ABL fusion shows that the expression is slightly increased. (Figure 12E). Additionally, a significant increase in expression was found in relapsed patients compared to patients with complete response. (Figure 12F). These results are consistent with the aforementioned and reinforce the importance of BCL-xL as a potential therapeutic target and biomarker.

This is consistent with that reported by Jin-Song, Yang et al. [90], where Kaplan Mayer and a multivariate analysis, showed that in colorectal cancer, high levels of BCL-XL expression have a poor survival compared to those patients who showed low expression (*p* = 0.016). Through a multivariate analysis, they suggest that overexpression of BCL-xL could serve as a prognostic marker for colorectal cancer. On the other hand, in adult acute myeloid leukemia a multivariate analysis showed that expression of BCL-XL was found as an independent negative prognostic factor for response to induction of therapy in patients older than 60 years with intermediate cytogenetic risk [91].

Finally, in multiple myeloma, using the dataset reported by Agnelli and Co. [92], in the analysis of the expression of BCL-xL according to the stage of the myeloma, a discrete but important increase in the expression of BCL-xL is found according to a more advanced stage (Figure 13A), which could suggest that the protein participates in the aggressiveness of the disease.

The dataset reported by Mulligan and Co. [93] is contrary to what is expected; an increased BCL-xL expression was found in living patients compared to deceased patients, according to the three-year total survival report (Figure 13B). This behavior was repeated in another dataset reported by Zhan and Co. [94], in which, although discreetly, a slight increase in the expression of BCL-xL was found in living patients compared to patients who died due to the disease. Interestingly, in the dataset reported by Raab and Co. [95], BCL-xL expression increased significantly on time dependent in MM.1S myeloma cell line after treatment with enzaustarin, a protein kinase C inhibitor with evidence of promoting apoptosis and inhibiting proliferation [70,95] (Figure 13C). These findings can open a new paradigm in the promotion of apoptosis in MM and its relationship with the anti-apoptotic protein BCL-xL.

All of these data and those reported in this review strongly suggest that BCL-XL could be a potential prognostic marker and the study as a therapeutic target is of great therapeutic expectation in hematological malignancies, as well as in other types of cancer.

## 4. Conclusions

Since the discovery and cloning of BCL-XL in 1993, more than 25 years ago, much important information has been accumulated revealing its importance in a large number of biological processes. As described in this review, BCL-XL appears to play an important role in the pathogenesis of some types of hematologic malignancies. This is due to its ability to control in cancer among other biological processes. In recent years, the Bcl-2 family of proteins has become an important therapeutic target in cancer. Therefore, in the coming years, the development of many therapies based on the regulation of this Bcl-2 family of protein (and specifically BCL-XL) will not be surprising.

## Figures and Tables

**Figure 1 ijms-23-02193-f001:**
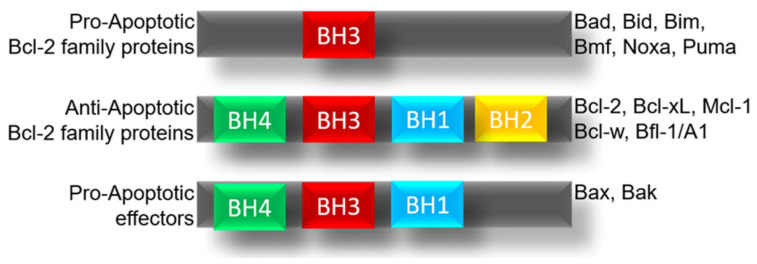
Members of the BCL-2 family and their role. Schematic representation of the members of BCL-2 family and his subunits.

**Figure 2 ijms-23-02193-f002:**
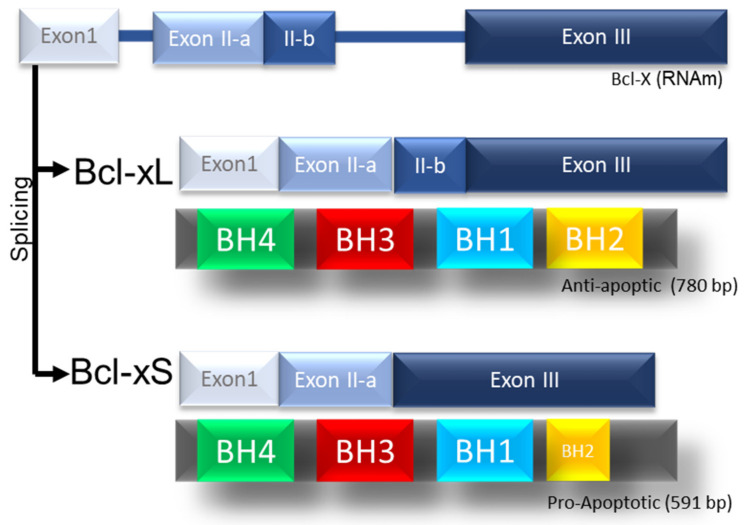
Scheme of the BCL-xL protein and its splicing products. Schematic representation of the splicing of BCL-2 and his products BCL-xL and Bcl-xS.

**Figure 3 ijms-23-02193-f003:**
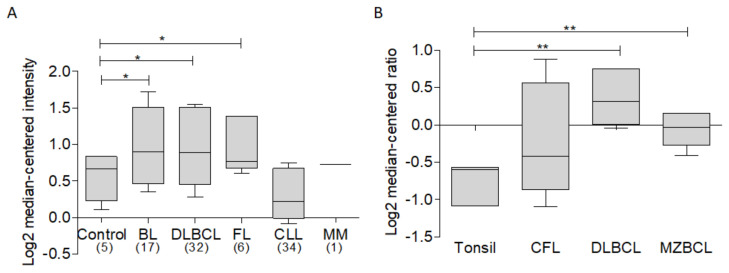
Analysis of BCL-xL expression in hematological malignancies. Bioinformatic analysis of BCL-xL expression was revised in ONCOMINE data base using different datasets (* *p* < 0.05). (**A**) The analysis of Basso et al. [29] of BCL-xL expression in different hematological malignancies including subtypes of lymphoma, leukemia, and multiple myeloma. (**B**) In Storz et al. [30], we observe higher expression levels of BCL-xL expression in DLBCL and MZBCL vs. tonsil used as a control (** *p* < 0.01).

**Figure 4 ijms-23-02193-f004:**
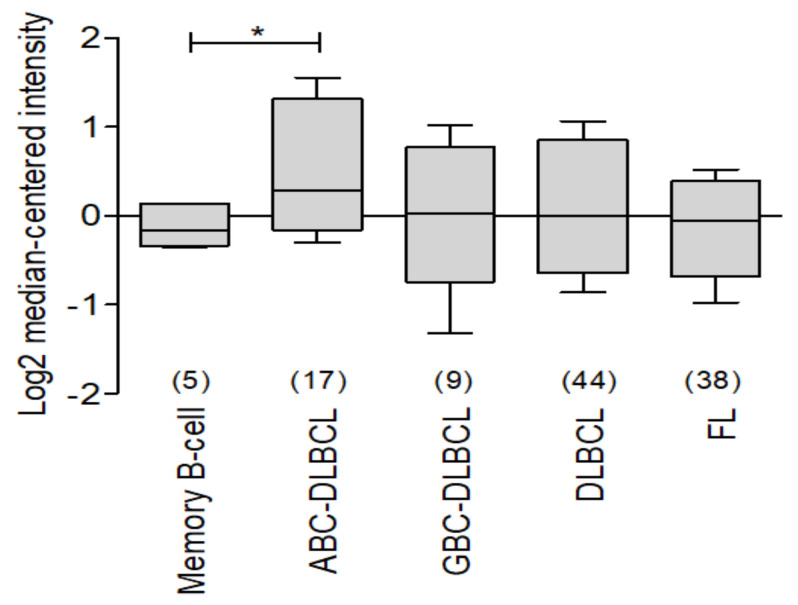
Analysis of BCL-xL expression in different subtypes of non-Hodgkin’s lymphoma. ONCOMINE BCL-xL expression was revised in Compagno Lymphoma [35]. An important expression of BCL-XL was observed in different subtypes of lymphoma versus memory B-cell used as control (* *p* < 0.05).

**Figure 5 ijms-23-02193-f005:**
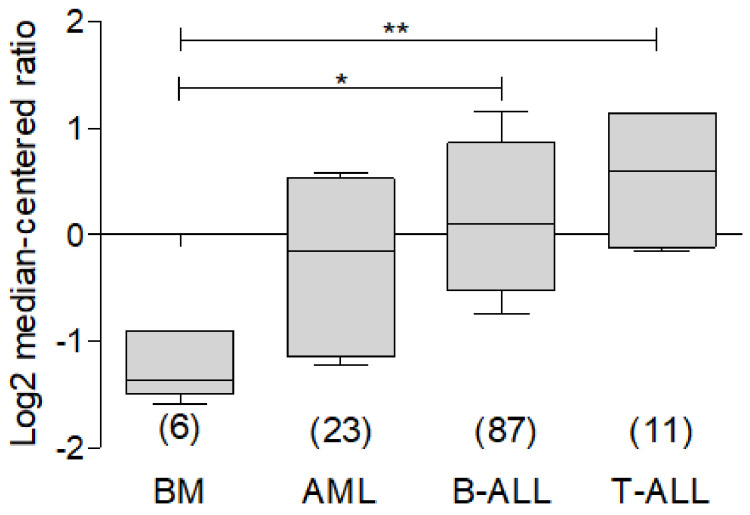
Analysis of BCL-xL expression in different subtypes of leukemia. Bioinformatic analysis with ONCMOINE of BCL-xL expression on leukemia. In the dataset of Anderson et al. [44], analysis of BCL-xL expression shows a significant higher expression in B-ALL and T-ALL compared to bone marrow (* *p* < 0.05, ** *p* < 0.001).

**Figure 6 ijms-23-02193-f006:**
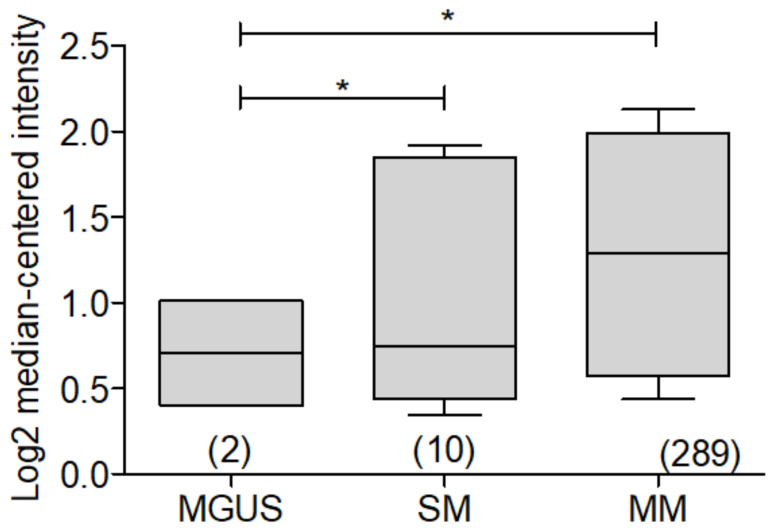
Analysis of BCL-xL expression in different myeloma subtypes. Bioinformatic analysis with ONCOMINE of BCL-xL expression in Myeloma was revised. The dataset analysis of Chapman et al. [4] of BCL-xL expression in MGUS, SM, and MM was performed, and the expression was found to be higher in MGUS (* *p* < 0.05).

**Figure 7 ijms-23-02193-f007:**
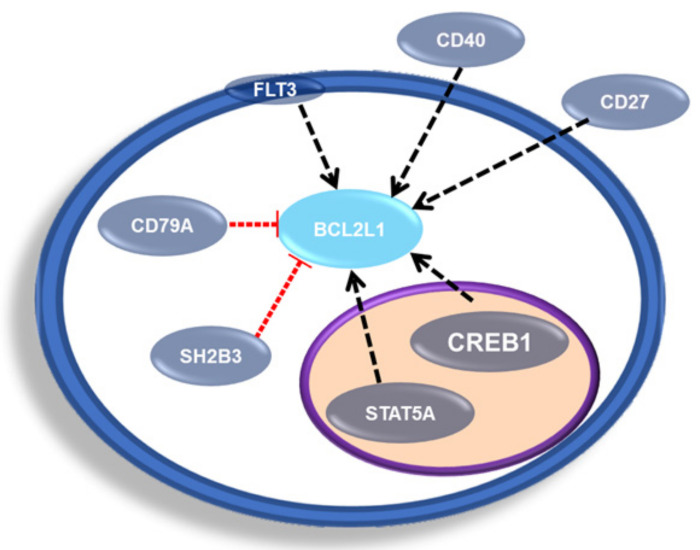
Representative scheme of BCL-xL transcriptional regulators. Schematic representation of transcription factors and proteins involved in the regulation of BCL2L1 transcription.

**Figure 8 ijms-23-02193-f008:**
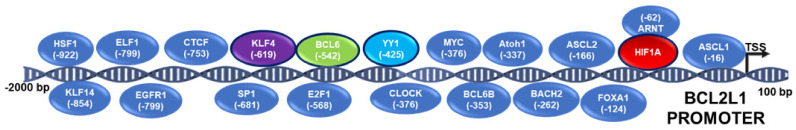
Scheme of transcription factors with the potential to regulate BCL-xL. Bioinformatic prediction of transcription factors potentially involved in the regulation of the BCL2L1 promotor.

**Figure 9 ijms-23-02193-f009:**
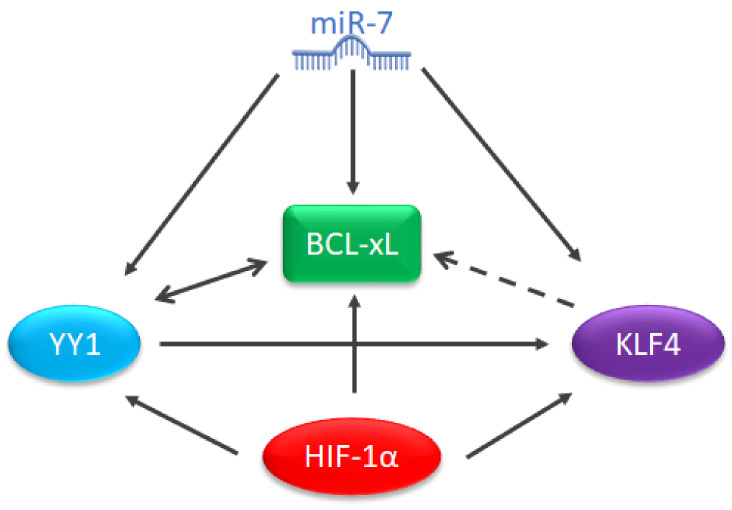
Proposal for a potential regulatory environment for BCL-xL. Schematic proposal or a regulatory environment of BCL-xL.

**Figure 10 ijms-23-02193-f010:**
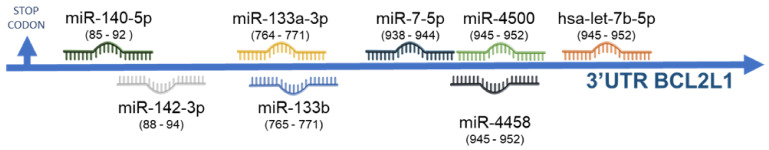
Scheme of microRNAs capable of regulating BCL-xL. Predictive analysis. Schematic representation of a bioinformatic prediction of microRNAs involved in the regulation of BCL2L1 by union to 3′UTR.

**Figure 11 ijms-23-02193-f011:**
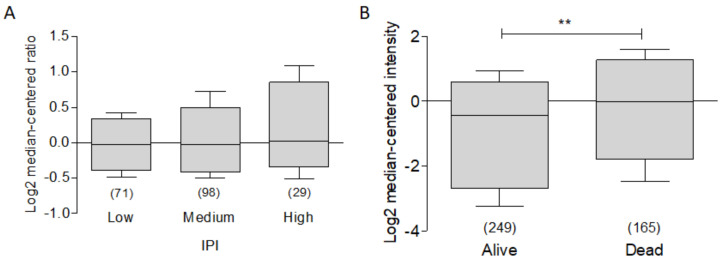
BCL-xL as a potential biomarker in Lymphoma. ONCOMINE BCL-xL expression was revised in Rosendwald et al. [86] for lymphoma. (**A**) There is a relative higher expression of BCL-xL and IPI value. (**B**) The dataset analysis of Lenz et al. [87] shows a non-significantly higher expression of BCL-xL in dead vs. alive patients.

**Figure 12 ijms-23-02193-f012:**
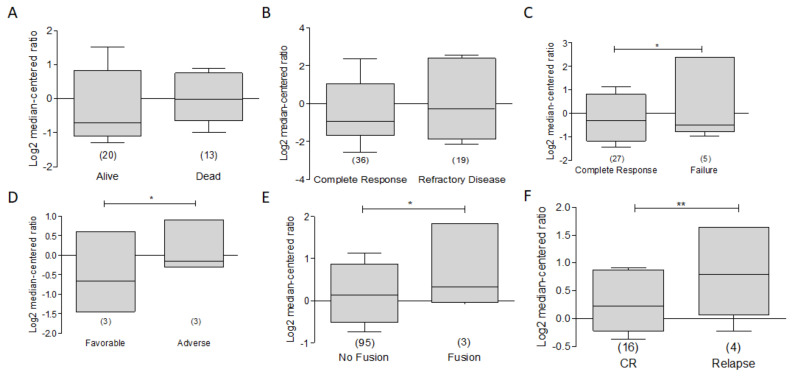
BCL-XL as a potential biomarker in Leukemia. ONCOMINE analysis of BCL-XL was revised and compared to clinical features of patients with Leukemia. (**A**) The dataset from the analysis conducted by Heuser [88] of BCL-XL expression was performed. A discrete increase was observed in dead patients. (**B**) In Bullinger’ leukemia dataset [89], a lower expression of BCL-xL was observed in patients with complete response vs. refractory disease. (**C**) Data from the study by Heuser is shown on responder vs. failure response patients (* *p* < 0.05). (**D**) In the same dataset lower expression of BCL-XL was related with a favorable outcome (* *p* < 0.05). (**E**) In the same dataset, the analysis of BCL-XL as a function of BCR-ABL is shown. (**F**) The expression of BCL-XL is shown in relapse patient vs. complete response patients (** *p* < 0.001).

**Figure 13 ijms-23-02193-f013:**
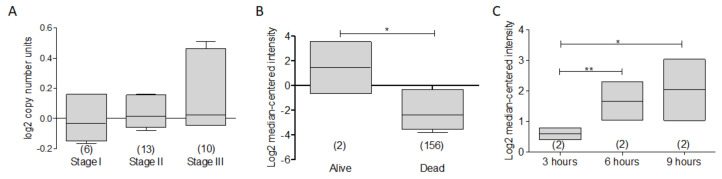
BCL-XL as a potential biomarker in Multiple Myeloma. (**A**) According to the stage of Myeloma, a discrete increase in the expression of BCL-XL is found in advanced stages. (**B**) In the dataset of Mullingan et al. [93], the expression of BCL-xL was higher in alive patients compared to dead (* *p* < 0.05). (**C**) In Raab et al. [95], there was a significant increase in the expression of BCL-XL in function of time (* *p* < 0.001, ** *p* < 0.05).

**Table 1 ijms-23-02193-t001:** Potential binding sites in BCL-xL promoter.

Transcription Factor	Potential Sites
ARNT::HIF1A	−62, −377, −532, −842
ASCL1	−16, −17, −112, −342, −383, −522, −523, −628, −835, −867
Ahr::Arnt	−62, −842
Ar	−96, −670, −856
Arnt	−377, −532
Arntl	−377
Ascl2	−166, −383, −523
Atoh1	−337
BACH2	63, −262
BCL6B	−353, −362, −540
Bcl6	−542, −734
CLOCK	−337, −376, −377, −532, −533
CTCF	70, −36, −83, −112, −293, −336, −574, −577, −645, −705, −710, −753
E2F1	−435, −568, −569, −715, −716, −944
EGR1	34, −49, −506, −799
ELF1	60, 22, −37, −488, −538, −799
FOXA1	−124
HSF1	−922
IRF1	−140
KLF4	−64, −175, −459, −619, −898
KLF14	37, −22, −46, −64, −79, −294, −602, −681, −714, −717, −738, −749, −763, −827, −854
MYC	−376, −533
SP1	20, −45, −78, −181, −247, −285, −412, −460, −492, −591, −620, −681, −704, −739, −748, −762, −803, −855
YY1	−425

**Table 2 ijms-23-02193-t002:** MicroRNAs related to hematologic malignancies: DLBCL, diffuse large B-cell lymphoma; FL, follicular lymphoma; AML, acute myeloid leukemia; ALL, acute lymphocytic leukemia; CLL, chronic lymphocytic leukemia; CML, chronic myeloid leukemia; MM, multiple myeloma. Modified from Lawrie CH. MicroRNA in hematological malignancies [66].

Hematological Malignancies	miRNAs Related
B-Cell Lymphoma	miR-17-92, miR-34a, miR-21.
DLBCL	miR-155, miR-125,
FL	miR-17-92,
AML	miR-155, miR-125, miR-181, miR-221/222, miR-29a, miR-124a.
ALL	miR-17-92, miR-125, miR-15a/16-1, miR-124a.
CLL	miR-181, miR-221/222, miR-29a, miR-15a/16-1.
CML	miR-17-92,
MM	miR-17-92, miR-15a/16-1, miR-21

**Table 3 ijms-23-02193-t003:** microRNAs that have been reported as possible panel of micro-markers on different subtypes of leukemia.

LeukermiRs	Leukemia Type	Expression	Role
miR-511, miR-222, miR-34	ALL	High	Leukemogenesis
miR-199a-3p, miR-223, miR-221, miR-26a.	ALL	Down	Leukemogenesis
miR-181b-5p, miR-181a-3p, miR-181a-5p, miR-342-3p	--	Dysregulated	Differentiate subtypes of leukemia.
miR-450a-5p, miR-1225a,	--	Dysregulated	Dysregulated in all subtypes of leukemia
miR-128, miR-146a, miR-155, miR-181a, miR-195	ALL	High	High expression vs. healthy control.
Let-7, miR-223	ALL	Lower	Lower expression in ALL vs. AML
miR-29c-5p	ALL	High	Increased expression allows one to difference between B and T ALL.

**Table 4 ijms-23-02193-t004:** lymphomiRs reported to be involved in lymphomagenesis.

LymphomiR	Target	Roll	Refs.
miR-34	FOXP3, SIRT1	Media transition pro-B a pre-B	[70]
miR-7	KLF4, YY1	Chemoresistance & migration	[61]
miR-17-5p	E2F1	Cell Cycle Regulation	[71]
miR-106b	P21/CDK1	Cell Cycle Regulation	[72]
miR-17-92 cluster	HIF-1a	Chemoresistance	[73]
miR-155	TP53INP	Tumor growth	[74]

**Table 5 ijms-23-02193-t005:** lymphomiRs reported on different types and subtypes of lymphoma.

Upregulated in DLBCL by small-RNA-seq [65]	Downregulated in DLBCL by small-RNA-seq [65]	Downregulated in DLBCL and BL [41]	Upregulated in FL [63]		Downregulated in FL [63]	
miR-124	miR-425	miR-150	miR-193a-5p	miR-1295	miR-17 *	miR-222
miR-532-5p	miR-141	miR-189	miR-193b *	miR-1471	miR-30a	miR-301b
	miR-145	miR-223	miR-345		miR-33a	miR-431 *
	miR-197	miR-768-3p	miR-513b		miR-106a *	
	miR-345	miR-15	miR-574-3p		miR-141	
	miR-424		miR-584		miR-202	
	miR-128		miR-663		miR-205	
	miR-122		miR-1287		miR-570	

* passenger strand.

**Table 6 ijms-23-02193-t006:** MicroRNAs with potential binding sites in the 3’UTR region of BCL-xL.

Target and microRNA	Position
Position 85–92 of BCL2L1 3’ UTR hsa-miR-140-5p	5’ ...CAUUGCCACCAGGAG--AACCACUA... ||| ||||||| 3’ GAUGGUAUCCCAUUUUGGUGAC
Position 88–94 of BCL2L1 3’ UTR hsa-miR-142-3p.1	5’ ...UGCCACCAGGAGAACCACUACAU... ||||||| 3’ AGGUAUUUCAUCCUUUGUGAUGU
Position 764–771 of BCL2L1 3’ UTR hsa-miR-133a-3p.1	5’ ...CCAUGACCAUACUGAGGGACCAA... ||||||| 3’ GUCGACCAACUUCCCCUGGUU
Position 765–771 of BCL2L1 3’ UTR hsa-miR-133a-3p.2	5’ ...CAUGACCAUACUGAGGGACCAAC... ||||||| 3’ GUCGACCAACUUCCCCUGGUUU
Position 765–771 of BCL2L1 3’ UTR hsa-miR-133b	5’ ...CAUGACCAUACUGAGGGACCAAC... ||||||| 3’ AUCGACCAACUUCCCCUGGUUU
Position 938–944 of BCL2L1 3’ UTR hsa-miR-7-5p	5’ ...UAUGGGAGCCCCAGGGUCUUCCC... ||||||| 3’ UGUUGUUUUAGUGAUCAGAAGGU
Position 945–952 of BCL2L1 3’ UTR hsa-miR-4458	5’ ...GCCCCAGGGUCUUCC----CUACCUCA... |||||| ||||||| 3’ AAGAAGGUGUGGAUGGAGA
Position 945–952 of BCL2L1 3’ UTR hsa-miR-4500	5’ ...GCCCCAGGGUCUUCCCUACCUCA... ||||||| 3’ UUCUUUGAUGAUGGAGU
Position 945–952 of BCL2L1 3’ UTR hsa-let-7b-5p	5’ ...GCCCCAGGGUCUUCCCUACCUCA... ||||||| 3’ UUGGUGUGUUGGAUGAUGGAGU

## Data Availability

Not applicable.

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
