# Peer review of "Roles and Regulation of BCL-xL in Hematological Malignancies"

_ijms, 2022, doi:10.3390/ijms23042193_

Round 1
Reviewer 1 Report
The work "Roles and regulation of BCL-xL in hematological malignancies"
of Morales-Martinez and Vega is great of interest and it is well organized, explaining in depth the role of BCL-xL and its regulation in several hematological malignancies. I suggest a correction before the publication:
Lines 87-91, page 3 of 25: please change the punctuation in the sentence
"such is the case of Venetoclax (ABT-199)(23), Navitoclax (ABT-263) (24). a small molecule inhibitor directed against BCL-2, clinical approved for use in ALL, CLL, and AML.(23)" in "such is the case of Venetoclax (ABT-199)(23) and Navitoclax (ABT-263) (24), a small molecule inhibitor directed against BCL-2, clinical approved for use in ALL, CLL, and AML.(23)
Author Response
The authors appreciate the important comments of the reviewer, and the modifications were made accordingly:
Lines 87-91, page 3 of 25: please change the punctuation in the sentence
"such is the case of Venetoclax (ABT-199)(23), Navitoclax (ABT-263) (24). a small molecule inhibitor directed against BCL-2, clinical approved for use in ALL, CLL, and AML.(23)" in "such is the case of Venetoclax (ABT-199)(23) and Navitoclax (ABT-263) (24), a small molecule inhibitor directed against BCL-2, clinical approved for use in ALL, CLL, and AML.(23)
R= The punctuation in the sentence was changed as suggested (Lines 93-95)
Reviewer 2 Report
The manuscript summarizes recent advances in its transcrip-23 tional regulation and the advances regarding the activities of BCL-XL in hematological malignancies, its possible role as a biomarker, and its possible clinical relevance in 25 malignancies.
The cited references are up-to-date, the review is well written.
Notes:
Figure 2: Please align the Exon III box should be allined with the other boxes
Figures 3, 4, 5, 6: The reference number of the source of figures is missing from the figure legend
Figure legends: abbreviations used on the figures are missing
Tables are not well structured. The notes for the tables should be under the table. Sturcture and style of Tables should follow the instructions.
Author Response
The authors appreciate the important comments of the reviewer, and the modifications were made accordingly:
Figure 2: Please align the Exon III box should be aligned with the other boxes
R= Figure 2 was modified according to the reviewer's suggestions (Line 70)
Figures 3, 4, 5, 6: The reference number of the source of figures is missing from the figure legend
R= The corresponding references were added in the corresponding figure legends (Lines 123, 124, 150, 185, 226, 412, 413, 441, 443, 444, 468 and 469 respectively)
Figure legends: abbreviations used on the figures are missing
R= Added missing figure legend abbreviations in the abbreviations section (Lines 27-31)
Tables are not well structured. The notes for the tables should be under the table. Structure and style of Tables should follow the instructions.
R= The tables were adjusted and structured according to the instructions (Lines 259-260, 293-294, 313-314, 320-321, 328-329 and 362-363 respectively).
Reviewer 3 Report
This looks like a pretty comprehensive review of BCL-XL. I don't have much comments except that the font of tables should following guidelines. Thank you.
Author Response
The authors appreciate the important comments of the reviewer, and the modifications were made accordingly:
This looks like a pretty comprehensive review of BCL-XL. I don't have much comments except that the font of tables should following guidelines. Thank you.
R= The tables were adjusted and structured according to the instructions (Lines 259-260, 293-294, 313-314, 320-321, 328-329 and 362-363 respectively).